# Understanding Anomaly Detection with Deep Invertible Networks through Hierarchies of Distributions and Features

**Robin Tibor Schirrmeister**[*]
University Medical Center Freiburg
Bosch Center for Artificial Intelligence
robin.schirrmeister@uniklinik-freiburg.de

**Yuxuan Zhou**
Bosch Center for Artificial Intelligence
Yuxuan.Zhou@bosch.com

**Tonio Ball**
University Medical Center Freiburg
tonio.ball@uniklinik-freiburg.de

**Dan Zhang**
Bosch Center for Artificial Intelligence
Dan.Zhang2@bosch.com

## Abstract

Deep generative networks trained via maximum likelihood on a natural image dataset like CIFAR10 often assign high likelihoods to images from datasets with different objects (e.g., SVHN). We refine previous investigations of this failure at anomaly detection for invertible generative networks and provide a clear explanation of it as a combination of model bias and domain prior: Convolutional networks learn similar low-level feature distributions when trained on any natural image dataset and these low-level features dominate the likelihood. Hence, when the discriminative features between inliers and outliers are on a high-level, e.g., object shapes, anomaly detection becomes particularly challenging. To remove the negative impact of model bias and domain prior on detecting high-level differences, we propose two methods, first, using the log likelihood ratios of two identical models, one trained on the in-distribution data (e.g., CIFAR10) and the other one on a more general distribution of images (e.g., 80 Million Tiny Images). We also derive a novel outlier loss for the in-distribution network on samples from the more general distribution to further improve the performance. Secondly, using a multi-scale model like Glow, we show that low-level features are mainly captured at early scales. Therefore, using only the likelihood contribution of the final scale performs remarkably well for detecting high-level feature differences of the out-of-distribution and the in-distribution. This method is especially useful if one does not have access to a suitable general distribution. Overall, our methods achieve strong anomaly detection performance in the unsupervised setting, and only slightly underperform state-of-the-art classifier-based methods in the supervised setting. Code can be found at https://github.com/boschresearch/hierarchical_anomaly_detection.

## 1 Introduction

One line of work for anomaly detection - to detect if a given input is from the same distribution as the training data - uses the likelihoods provided by generative models. Through likelihood maximization,

---

[*]This work was partially done during an internship at the Bosch Center for Artificial Intelligence.

they are trained to yield high likelihoods on the in-distribution inputs (a.k.a. inliers).[2] After training, one may expect out-of-distribution inputs (a.k.a. outliers) to have lower likelihoods than the inliers. However, this is often not the case. For example, Nalisnick et al. [21] showed that generative models trained on CIFAR10 [14] assign higher likelihoods to SVHN [23] than to CIFAR10 images.

Several works have investigated a potential reason for this failure: The image likelihoods of deep generative networks can be well-predicted from simple factors. For example, the deep generative networks' image likelihoods highly correlate with: the image encoding sizes from a lossless compressor such as PNG [31]; background statistics, e.g., the number of zeros in Fashion-MNIST/MNIST images [26]; smoothness and size of the background [16]. These factors do not directly correspond to the type of object, hence the type of object does not affect the likelihood much.

In this work, we first synthesize these findings into the following hypothesis: A convolutional deep generative network trained on any image dataset learns low-level local feature relationships common to all images - such as smooth local patches - and these local features, forming the *domain prior*, dominate the likelihood. One can therefore expect a smoother dataset like SVHN to have higher likelihoods than a less smooth one like CIFAR10, irrespective of the image dataset the network was trained on. Following prior works, we take Glow networks [12] as the baseline model for our study.

Next, we report several new findings to support the hypothesis: (1) Using a fully-connected instead of a convolutional Glow network, likelihood-based anomaly detection works much better for Fashion-MNIST vs. MNIST, indicating a convolutional model bias. (2) Image likelihoods of Glow models trained on more general datasets, e.g., 80 Million Tiny Images (Tiny), have the highest average correlation with image likelihoods of models trained on other datasets, indicating a hierarchy of distributions from more general distributions (better for learning domain prior) to more specific distributions. (3) The likelihood contributions of the final scale of the Glow network correlate less between different Glow networks than the likelihood contributions of the earlier scales, while the overall likelihood is dominated by the earlier scales. This indicates a hierarchy of features inside the Glow network scales, from more generic low-level features that dominate the likelihood to more distribution-specific high-level features that are more informative about object categories.

Finally, leveraging the two novel views of a hierarchy of distributions and a hierarchy of features, we propose two likelihood-based anomaly detection methods. From the hierarchy-of-distributions view, we use likelihood ratios of two identical generative architectures (e.g., Glow), one trained on the in-distribution data (e.g., CIFAR10) and the other one on a more general distribution (e.g., 80 Million Tiny Images), refining previous likelihood-ratio-based methods. To further improve the performance, we additionally train our in-distribution model on samples from the general distribution using a novel outlier loss. From the hierarchy-of-features view, we show that using the likelihood contribution of the final scale of a multi-scale Glow model performs remarkably well for anomaly detection.

Our manuscript advances the understanding of the anomaly detection behavior of deep generative neural networks by synthesizing previous findings into two novel viewpoints accounting for the hierarchical nature of natural images. Based on this concept, we propose two new anomaly detection methods which reach strong performance, especially in the unsupervised setting. Our experiments are more extensive than previous likelihood-ratio-based methods on images, especially in the unsupervised setting, and therefore also fill an important empirical gap in the literature.

## 2   Common Low-Level Features Dominate the Model Likelihood

The following hypotheses synthesized from the findings of prior work motivate our methods to detect if an image is from a different object recognition dataset:

1. Distributions of low-level features form a domain prior valid for all natural images (see Fig. 1A).
2. These low-level features contribute the most to the overall likelihood assigned by a deep generative network to any image (see Fig. 1B and C).
3. How strongly which type of features contributes to the likelihood is influenced by the model bias (e.g., for convolutional networks, local features dominate) (see Fig. 1C).

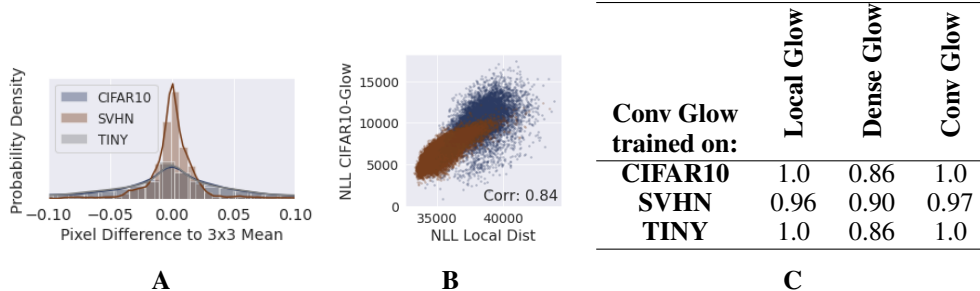

| Conv Glow trained on: | Local Glow | Dense Glow | Conv Glow |
|---|---|---|---|
| CIFAR10 | 1.0 | 0.86 | 1.0 |
| SVHN | 0.96 | 0.90 | 0.97 |
| TINY | 1.0 | 0.86 | 1.0 |

**A**      **B**      **C**

Figure 1: Low-level features and model bias. **A**: Distributions over local pixel value differences have overlapping high-density regions for Tiny, SVHN, CIFAR10 . **B**: Likelihoods extracted from the local pixel-difference distributions correlate with CIFAR10-Glow likelihoods. **C**: Likelihood correlations for different types of Glow networks trained on CIFAR10 with regular convolutional Glow networks trained on CIFAR10, SVHN and Tiny. Correlations are almost the same for convolutional Glow networks and local Glow networks trained on $8 \times 8$ patches. Correlations are smaller for fully-connected/dense Glow networks.

For the first hypothesis, we start from defining low-level features. They are features that can be extracted in few computational steps (these will be local features for convolutional models like Glow). As an example, we use the difference of a pixel value to the mean of its $3 \times 3$ neighbouring pixels. As natural images are smooth, the distributions over such per-pixel difference of SVHN, CIFAR10 and 80 Million Tiny Image (Tiny) depicted in Fig. 1A are all zero centered. Smoother images will produce smaller differences among neighbouring pixels, therefore the density of SVHN has the highest peak around zero. A significant overlapping high-density regions for SVHN, CIFAR10 and Tiny show that these low-level features are common to natural images, and thus not useful for anomaly detection.

Next, we examine the second hypothesis by showing that low-level per-image likelihoods highly correlate with Glow network likelihoods. On the pixel level, we compute the pixel difference and estimate its density using a histogram with 100 equally distanced bins. Using the estimated density, we can get the conditional (on $3 \times 3$ neighbours) per-pixel likelihoods of each image. Summing the per-pixel likelihoods over the entire image, we obtain pixel-level per-image pseudo-likelihood, which is not a correct likelihood of image, but a proxy measure of low-level feature contributions to the image-level likelihood. The low-level pseudo-likelihoods of SVHN and CIFAR10 images have Spearman correlations[3] $> 0.83$ with likelihoods of Glow networks trained on CIFAR10 (see Fig 1B), SVHN or Tiny. We also trained small modified Glow networks on $8 \times 8$ patches cropped from the original image. These local Glow networks' likelihoods correlate even more with the full Glow networks' likelihoods ($> 0.95$), further suggesting low-level local features dominate the total likelihood (see Fig 1C). To validate that the low-level features dominating the likelihoods are independent of the semantic content of the image, we mix two images in Fourier space by combining the amplitudes of one image's Fourier transform with the phases of the other image's Fourier transform, and then evaluate the likelihood of the resulting image using the same pretrained Glow model (see supp. material S3 for details). The mixed images are semantically much more coherent with the images that provide the phase information, yet their Glow likelihoods correlate more strongly with the Glow likelihoods of images sharing the same amplitudes ($> 0.8$ vs. $< 0.05$).

Lastly, we show that what features are extracted and how much each feature contributes to the likelihood depend on the type of model. When training a modified Glow network that uses fully connected instead of convolutional blocks (see supp. material S2.2) on Fashion-MNIST and MNIST, the image likelihoods among them do not correlate (Spearman correlation $-0.2$). The fully-connected Fashion-MNIST network achieves worse likelihoods (4.8 vs. 2.9 bpd), but is much better at anomaly detection (81% AUROC for Fashion-MNIST vs. MNIST; 15% AUROC for convolutional Glow).

Consistent with non-distribution-specific low-level features dominating the likelihoods, we find full Glow networks trained independently on CIFAR10, CIFAR100, SVHN or Tiny produce highly correlated image likelihoods (Spearman correlation $> 0.96$ for all pairs of models, see Fig. 1 **C**). The same is true to a lesser degree for Fashion-MNIST and MNIST (Spearman correlation 0.85).

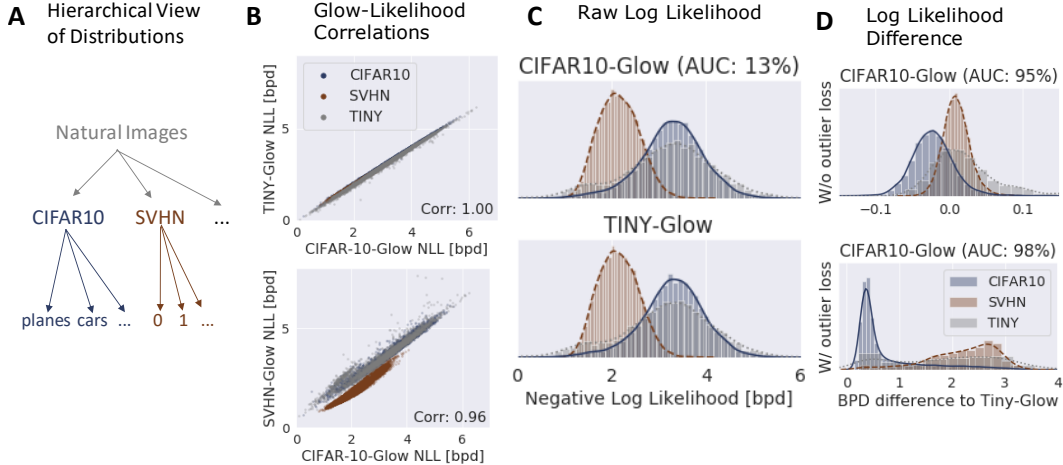

Figure 2: Overview of the hierarchy-of-distributions approach. **A**: Schematic hierarchical view of image distributions. To approximate the distribution of natural images, we use 80 Million Tiny images (Tiny). **B,C,D**: Results of Glow networks trained on CIFAR10, SVHN and Tiny. **B**: Likelihoods rank-correlate almost perfectly for the Glow networks trained on CIFAR10 and Tiny on all three datasets (top), while rank correlations remain very close to 1 for CIFAR10-Glow and SVHN-Glow (bottom), validating that the main likelihood contribution comes from the domain prior. **C**: Distribution plots show almost identical plots for CIFAR10 and Tiny-Glow and a low area under the receiver operating curve (AUC) for CIFAR10 vs. SVHN anomaly detection. **D**: In contrast, the log likelihood difference between CIFAR10-Glow and Tiny-Glow reaches substantially higher AUCs (top), further increased by using our outlier loss (bottom) (see Section 3.1).

Taken together, the evidence suggests convolutional generative networks have model biases that guide them to learn low-level feature distributions of natural images (domain prior) well, at the expense of anomaly detection performance. Based on this understanding, next we propose two methods to remove this influence of model bias and domain prior on likelihood-based anomaly detection.

## 3 Hierarchy of Distributions

The models trained on Tiny have the highest average likelihood across all evaluated datasets. This inspired us to use a hierarchy of distributions: CIFAR10 and SVHN are subdistributions of natural images, CIFAR10-planes are a subdistribution of CIFAR10, etc. (see Fig. 2).

We use this hierarchy of distributions to derive a log-likelihood-ratio-based anomaly detection method:

1. Train a generative network on a general image distribution like 80 Million Tiny Images;
2. Train another generative network on images drawn from the in-distribution, e.g., CIFAR10;
3. Use their likelihood ratio for anomaly detection.

Formally, given the general-distribution-network likelihood $p_g$ and the specific-in-distribution-network likelihood $p_{in}$, our anomaly detection score (low scores indicate outliers) is:

$$\log\left(\frac{p_{\text{in}}(x)}{p_{\text{g}}(x)}\right) = \log\left(p_{\text{in}}(x)\right) - \log(p_{\text{g}}(x)). \tag{1}$$

### 3.1 Outlier Loss

We also derive a novel outlier loss on samples $x_{\text{g}}$ from the more general distribution based on the two networks likelihoods. Concretely, we use the log-likelihood ratio after temperature scaling by $T$ as the logit for binary classification:

$$L_{\text{o}} = -\lambda \cdot \log\left(\sigma\left(\frac{\log(p_{\text{g}}(x_{\text{g}})) - \log(p_{\text{in}}(x_{\text{g}}))}{T}\right)\right) = -\lambda \cdot \log\left(\frac{\sqrt[T]{p_{\text{g}}(x_{\text{g}})}}{\sqrt[T]{p_{\text{in}}(x_{\text{g}})} + \sqrt[T]{p_{\text{g}}(x_{\text{g}})}}\right), \tag{2}$$

where $\sigma$ is the sigmoid function and $\lambda$ is a weighting factor.

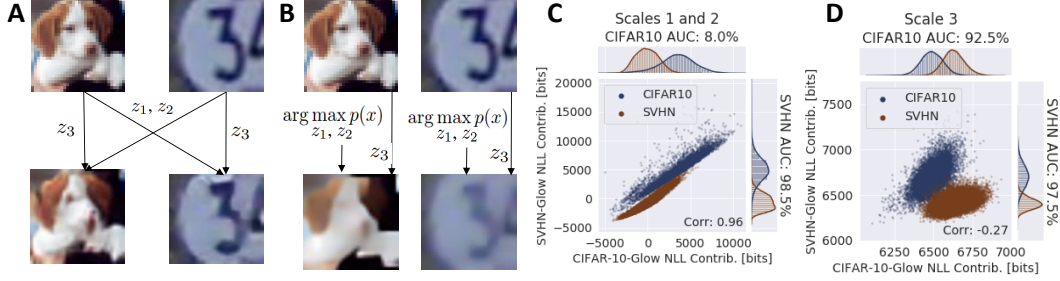

Figure 3: Overview of the hierarchy of features motivation. **A,B**: Showcasing features at different scales in the Glow-network. Top images are examples from CIFAR10 and SVHN. **A**: Bottom two images are obtained by mixing features in the Glow-network trained on CIFAR10 as follows. We compute the three scale outputs of the Glow-network $z_1$, $z_2$ and $z_3$, mix them between both images and invert again. For the bottom-left image, we take the earlier-scale features $z_1$ and $z_2$ from the SVHN image and the final-scale features $z_3$ from the CIFAR10 image and vice versa for the bottom-right image. Note the image class is completely determined by $z_3$. **B**: Images are optimized to maximize the CIFAR10-Glow-network likelihood $p(x)$ while keeping $z_3$ constant as follows. We keep $z_3$ from original Glow output fixed, and use the inverse pass of the network to optimize $z_1$ and $z_2$ via gradient ascent to maximize $\log(p(x))$. Only the global shape remains visible in the optimized images, while low-level structures have been blurred away. Such observation indicates that smoother local low-level features induce higher likelihood response of the model, once again confirming the strong influence of domain prior on the model likelihood. **C,D**: We use two Glow models trained on CIFAR10 and SVHN, respectively. The log likelihood they obtain on CIFAR10 and SVHN is split into log likelihood contributions of $z_1$, $z_2$ and $z_3$. The two plots show that (i) the summed contributions for $z_1$ and $z_2$ have very high rank correlation between both models (**C**); while the rank correlation drops for $z_3$ (**D**) and (ii) the range of the contributions is much larger for $z_1$ and $z_2$, showing that $z_1$ and $z_2$ dominate the total log likelihood.

## 3.2 Extension to the Supervised Setting

In all previous parts, our method is presented in an unsupervised setting, where the labels of the inliers are unavailable. We extend our method to the supervised setting with two main changes in our training. First, the Glow model $p_{\mathrm{in}}(x)$ uses a mixture of Gaussians for the latent $z$, i.e., each class corresponding to one mode. Second, the outlier loss is extended for each mode of $p_{\mathrm{in}}(x)$ by treating samples from the other classes as the negative samples, i.e., the same as $\{x_{\mathrm{g}}\}$ in Eq. 2.

## 4 Hierarchy of Features

The image likelihood correlations between models trained on different datasets reduce substantially when evaluating the likelihood contributions of the final scale of the Glow network (see Fig. 3). Here, the adopted Glow network has three scales. At the first two scales, i.e., $i = 1$ and 2, the layer output is split into two parts $h_i$ and $z_i$, where $h_i$ is passed onto the next scale and $z_i$ is output as one part of the latent code $z$. The output at the last scale is $z_3$, which together with $z_1$ and $z_2$ makes up the complete latent code $z$. In terms of $y_1 = (h_1, z_1)$, $y_2 = (h_2, z_2)$, $y_3 = z_3$ and $h_0 = x$, the logarithm of the learned density $p(x)$ can be decomposed into per-scale likelihood contributions $c_i(x)$ as

$$\log p(x) = \sum_i c_i(x) = \sum_i \log p_{\mathrm{z}}(z_i) + \log \left| \det \left( \frac{\partial y_i}{\partial h_{i-1}} \right) \right|. \tag{3}$$

The log-likelihood contributions $c_3(x)$ of the final third scale of Glow networks correlate substantially less than the full likelihoods for Glow networks trained on the different datasets (0.26 mean correlation vs. 0.99 mean correlation). This is consistent with the observation that last-scale dimensions encode more global object-specific features (see Fig. 3 and [4]). Therefore, we use $c_3(x)$ as our anomaly detection score (low scores indicate outliers).

Note that, here we do not condition $z_1$ on $z_2$ or $z_2$ on $z_3$, whereas other implementations often make $z_1$ dependent of $z_2$ such as $z_1 \sim N(f(z_2), g(z_2)^2)$ with $f, g$ being small neural networks. Such

Table 1: Anomaly detection performance (AUCs in %) of using the log-likelihood ratio of the in-distribution and general distribution model. For the general distribution models, we have three options, i.e., the general-purpose image compressor PNG [31] plus Tiny-Glow and Tiny-PCNN respectively trained on Tiny Images. Here, the Glow and PCNN trained on Tiny Images are also used as the starting point to train the in-distribution models. In the supp. material, we compare the results with training the two in-distribution models from scratch.

| In-dist. | OOD | Glow (in-dist.) diff to: | | | | PCNN (in-dist.) diff to: | | | |
|---|---|---|---|---|---|---|---|---|---|
| | | None | PNG | Tiny-Glow | Tiny-PCNN | None | PNG | Tiny-Glow | Tiny-PCNN |
| **SVHN** | CIFAR10 | 98.3 | 74.4 | **100.0** | **100.0** | 97.9 | 76.8 | **100.0** | **100.0** |
| | CIFAR100 | 97.9 | 79.5 | **100.0** | **100.0** | 97.4 | 81.3 | **100.0** | **100.0** |
| | LSUN | 99.6 | 96.8 | **100.0** | **100.0** | 99.4 | 98.1 | **100.0** | **100.0** |
| **CIFAR10** | SVHN | 8.8 | 75.4 | **93.9** | 16.6 | 12.6 | 82.3 | **94.8** | 94.4 |
| | CIFAR100 | 51.7 | 57.3 | **66.8** | 53.4 | 51.7 | 57.1 | 57.5 | **63.5** |
| | LSUN | 69.3 | 83.6 | **89.2** | 16.8 | 74.8 | 87.6 | **93.6** | 92.9 |
| **CIFAR100** | SVHN | 10.3 | 68.4 | **87.4** | 18.3 | 13.7 | 76.4 | **91.3** | 90.0 |
| | CIFAR10 | 49.2 | 44.1 | 52.8 | **54.2** | 49.1 | 44.2 | 48.3 | **54.5** |
| | LSUN | 66.3 | 77.5 | **81.0** | 19.1 | 71.7 | 82.7 | **90.0** | 87.6 |
| | Mean | 61.3 | 73.0 | **85.7** | 53.2 | 63.2 | 76.3 | 86.2 | **87.0** |

dependency is removable by transforming to $z_1' = \frac{(z_1 - f(z_2))}{g(z_2)}$, with $z_1'$ now independent of $z_2$ as $z_1' \sim N(0,1)$, and this type of transformation can already be learned by an affine coupling layer applied to $z_1$ and $z_2$, hence the explicit conditioning of other implementations does not fundamentally change network expressiveness. We do not use it here and do not observe bits/dim differences between our implementation and those that use it (see supp. material S7.1 for details).

# 5 Experiments

For the main experiments, we use SVHN [23], CIFAR10 [14], CIFAR100 [15] as inlier datasets and use the same and LSUN [36] as outlier datasets. Results for further outlier datasets can be found in the supp. material S7.4. 80 Million Tiny Images [34] serve as our general distribution dataset in the log likelihood-ratio based anomaly detection experiments and is also used in the outlier loss as given in Eq. 2 when training two generative models, i.e., Glow and PixelCNN++ (PCNN) (see supp. material S4 for their training details). In-distribution Glow and PixelCNN++ models are finetuned from the models pre-trained on Tiny for more rapid training, see supp. material S7.2 for details and ablation studies. Our reported results are averaged over 3 random seeds.

## 5.1 Anomaly Detection based on Log-Likelihood Ratio

In Tab. 1, we compare the raw log-likelihood based anomaly detection (i.e., **diff to: None**) with the log-likelihood ratio based ones (i.e., **diff to: PNG, Tiny-Glow, Tiny-PCNN**). The raw log-likelihood based scheme underperforms the log-likelihood ratio based ones that use Tiny-Glow and Tiny-PCNN. However, when using PNG to remove the domain prior as proposed in [31][4], it sometimes performs worse than the raw log-likelihood based scheme, e.g., SVHN as the in-distribution vs. the other three OODs. This relates to the remaining model bias, as Glow trained on the in-distribution encodes the domain prior differently to PNG. Also note that using PCNN as the general-distribution model for Glow does not work well for CIFAR10/100. This is because Tiny-PCNN has very large bpd gains over Glow for the CIFAR-datasets and less large gains for SVHN. On average across datasets, it works best to use matching general and in-distribution models (Glow for Glow, PCNN for PCNN), validating our idea of a model bias. Also note that our SVHN vs. CIFAR10 results already outperform the likelihood-ratio-based results of Ren et al. [26] slightly (93.9% vs. 93.1% AUROC), and we observe further improvements with outlier loss in Section 5.3. Ren et al. [26] used a noised version of the in-distribution as the general distribution and only tested it on SVHN vs. CIFAR10. Comparing to Tiny, it is less representative as a domain prior, and thus its performance on more complicated datasets requires further assessment.

Table 2: Raw log-likelihoods vs. log-likelihood ratios, where CIFAR10 is the in-distribution.

| Out-dist | Scale | Raw | Diff |
|---|---|---|---|
| **SVHN** | Full | 8.8 | **93.9** |
| | $16 \times 16$ | 7.0 | 84.6 |
| | $8 \times 8$ | 13.5 | 48.9 |
| | $4 \times 4$ | *92.9* | 83.6 |
| **CIFAR100** | Full | 51.7 | **66.8** |
| | $16 \times 16$ | 50.7 | 55.7 |
| | $8 \times 8$ | 53.5 | 56.3 |
| | $4 \times 4$ | *60.0* | 66.1 |
| **LSUN** | Full | 69.3 | **89.2** |
| | $16 \times 16$ | 70.3 | 63.6 |
| | $8 \times 8$ | 56.5 | 74.0 |
| | $4 \times 4$ | *82.8* | 75.1 |

Table 3: Different outlier losses, where CIFAR10 is the in-distribution.

| Out-dist | Loss | Raw | 4x4 | Diff |
|---|---|---|---|---|
| **SVHN** | None | 8.8 | 92.9 | 93.9 |
| | Margin | 84.2 | 84.2 | 96.5 |
| | Ours | 95.5 | 96.4 | **98.6** |
| **CIFAR100** | None | 51.7 | 60.0 | 66.8 |
| | Margin | 72.3 | 71.7 | 71.1 |
| | Ours | 84.9 | **85.4** | 84.5 |
| **LSUN** | None | 69.3 | 82.8 | 89.2 |
| | Margin | 82.0 | 82.0 | 75.7 |
| | Ours | 94.9 | **95.1** | 94.1 |

**Medical Dataset** To further validate our log-likelihood ratio approach on a different domain, we setup an experiment on the medical BRATS Magnetic Resonance Imaging (MRI) dataset. We use one MRI modality as in-distribution and the other three as OOD. The raw likelihood, the log-likelihood ratio to Tiny-Glow and to BRATS-Glow (trained on all modalities) yield AUROC 53.3%, 68.3% and 78.3%, respectively. So, Tiny also serves as a general distribution for the very different medical images, and a distribution from the more specific domain further improves the performance.

The log-likelihood ratio approach can likely be applied to more than images. In the above, we have already shown the application to typical image datasets and, without adaptation, to medical MRI images. In the text/NLP domain, it may be used with Wikitext-2 as the general dataset, since Wikitext-2 already worked well as an outlier dataset in [8]. In the audio domain, the domain prior may come from strong dependencies of the signal values on short timescales, similar to the smoothness of natural images. If a suitable general dataset needs to be created, it does not require labels and may even profit from noisy/unclean data. Therefore there is no principal obstacle preventing collection of such data, including concatenating existing datasets.

## 5.2 Anomaly Detection based on Last-scale Log-likelihood Contribution

As an alternative to remove the domain prior by using, e.g., Tiny-Glow, our hierarchy-of-features view suggests to use the log-likelihood contributed by the high-level features attained at the last scale of the Glow model. It is orthogonal to the log-likelihood ratio based scheme, and can be used when the general distribution is unavailable. As shown in Tab. 2, using the raw log-likelihood on the last scale ($4 \times 4$), consistently outperforms the conventional log-likelihood comparison on the full scale, but performs slightly worse than using the log-likelihood ratio in the full scale. Note that we don't expect the log-likelihood ratio on the last scale to be the top performer, as the domain prior is mainly reflected by the earlier two scales, see Fig. 3.

## 5.3 Outlier Losses

When training the in-distribution model, we can use the images from Tiny as the outliers to improve the training. Tab. 3 shows that our outlier loss as formulated in Eq. 2 consistently outperforms the margin loss [8] when combining with three different types of log-likelihood based schemes, i.e., raw log-likelihood, raw log-likelihood at the last scale $4 \times 4$ and log-likelihood ratio. We note that as the margin loss leads to substantially less stable training than our loss, see the supp. material S7.5.

We also experiment on adding the outlier loss to the training loss of Tiny-Glow, i.e., using the in-distribution samples as outliers. This further improves the anomaly detection performance, see **Diff**† of Tab. 4, while **Diff** only uses the outlier loss for training the in-distribution Glow-network.

## 5.4 Unsupervised vs. Supervised Setting

From the unsupervised to the supervised setting, Tab. 4 further reports the numbers achieved by using the class-conditional in-distribution Glow-network and treating inputs from other classes as outliers. We observe further improved anomaly detection performance.

Table 4: Anomaly detection performance summary (AUC in %). The new term **Diff**† means to use in-distribution samples as the outliers to train Tiny-Glow, see Sec. 5.4. **OE**, proposed by Hendrycks et al. [8], stands for margin-based outlier loss for PixelCNN++, **MSP-OE** from the same work stands for entropy of classifier predictions with entropy outlier loss.

| | Setting | Unsupervised | | | | Supervised | | | |
|---|---|---|---|---|---|---|---|---|---|
| **In-dist** | **Out-dist** | **4×4** | **Diff** | **Diff†** | **OE** | **4×4** | **Diff** | **Diff†** | **MSP-OE** |
| | SVHN | 96.4 | 98.6 | **99.0** | 75.8 | 96.1 | 98.6 | **99.1** | 98.4 |
| **CIFAR10** | CIFAR100 | 85.4 | 84.5 | **86.8** | 68.5 | 88.3 | 87.4 | 88.5 | **93.3** |
| | LSUN | 95.1 | 94.1 | **95.8** | 90.9 | 95.3 | 94.1 | 96.2 | **97.6** |
| | Mean | 92.3 | 92.4 | **93.8** | 78.4 | 93.3 | 93.4 | 94.6 | **96.4** |
| | SVHN | 84.5 | 82.2 | **85.4** | - | **89.6** | 88.6 | 89.4 | 86.9 |
| **CIFAR100** | CIFAR10 | 61.9 | 59.8 | **62.5** | - | 67.0 | 64.9 | 65.3 | **75.7** |
| | LSUN | 84.6 | 82.4 | **85.4** | - | 85.7 | 84.3 | **86.3** | 83.4 |
| | Mean | 77.0 | 74.8 | **77.8** | - | 80.8 | 79.3 | 80.3 | **82.0** |
| | Mean | 84.7 | 83.6 | **85.8** | - | 87.0 | 86.3 | 87.5 | **89.2** |

Overall, our approach only slightly underperforms the approach MSP-OE [8] with inlier class labels (**Supervised**), while being substantially better without inlier class labels (**Unsupervised**), see Tab. 4. In contrast to observations by Hendrycks et al. [8] for their unsupervised setup, we do not experience a severe degradation of the anomaly detection performance from the lack of class labels.

# 6   Related Work

We present an overview over anomaly detection approaches with a focus on recent work closely related to the ideas of a hierarchy of distributions and a hierarchy of features.

**Classifier-based Methods** Multi-class classifiers trained to discriminate in-distribution classes have been used for anomaly detection. Hendrycks and Gimpel [7] used the maximum softmax response as the score of normality. Different data augmentation schemes [19, 17, 8, 9] further enforced its performance. Lee et al. [18] alternatively modeled the class-conditional features attained by the hidden layers of the classifier as multivariate Gaussians, and then used the Mahalanobis distance of Gaussians for anomaly detection. Another recent work [6] used the gradient norm of the log-sum-exp of the class logits over the input for anomaly detection. In the context of self-supervised learning, class becomes the type of transformations [2, 5]. Self-supervised contrastive training improved the anomaly detection performance of multi-class classifiers [35].

Instead of exploiting multi-class classifiers, a different approach is to train a one-class classifier to directly discriminate inliers and outliers. One-class support vector machines are trained to return positive values only in a small region containing the inliers and negative values elsewhere [30]. This approach has also been used in forming the latent space of deep autoencoders [27, 28, 29]. Ruff et al. [29] also used samples from a general distribution as outliers. Steinwart et al. [33] drawn outliers from uniform distribution. In the supp. material S9, we also report results using a in-distribution-vs-general-distribution classifier.

**Reconstruction-based Methods** Another line of work is to learn the features and generation of inliers by reconstructing the training samples either in their input space or latent space, e.g., [25, 24, 1, 11]. At test time, an outlier is then detected if reconstruction is poor. However, owing to large capacity of deep neural networks, reconstruction loss alone may not be a reliable metric for anomaly detection. Huang et al. [10] proposed to additionally use the joint likelihood of latent variables, which is obtained by using a neural rendering model to invert multi-class CNN-based classifier.

**Input Likelihood-based Methods** Generative modeling through maximum likelihood estimation tries to enforce high likelihoods on inliers. Under the normalization constraint, the likelihoods of outliers are expected to be low (ideally zero). However, their anomaly detection performance is often unsatisfactory [32, 21, 8]. Outliers may attain even higher likelihoods than inliers. Recent work [22] related the poor performance to sampling in a high-dimensional space, namely, inliers being mapped to the typical set of the latent code rather than the high likelihood area. They proposed to address this issue by batch-wise anomaly detection, whose application is more limited than instance-wise anomaly detection. A different approach [3] combined input likelihoods with inlier classifiers. Che

et al. [3] trained a class-conditional generative model with an auxiliary adversarial loss to disentangle the class information from the rest latent representation. The achieved performance is better than ours in the supervised setting, while our methods mainly target and work in the unsupervised setting. It can be interesting to exploit their way of incorporating the label information into our model training.

Our work is also input likelihood-based. Our analysis in Section 2 showed that convolutional networks trained on one natural image dataset will learn low-level feature distribution that is common to the whole domain, and such domain prior dominates the likelihood. The concurrent work [13] found results consistent with ours. We further exploited hierarchies of distributions or hierarchies of features as explained in Section 3 and 4 to improve the anomaly detection performance.

*Hierarchy of Distributions*: Our hierarchy-of-distributions likelihood-ratio method relates to prior [26] and concurrent [31] methods as follows. Ren et al. [26] used a noised version of the in-distribution as the general distribution. Their method always requires training of two models for each in-distribution, our method has the option of only require one training of a general distribution model which we can reuse for any new in-distribution as long as it is as a subdistribution of the general distribution. In our method, the challenge is to find a suitable general distribution, while in their method the challenge is to find a suitable noise model. In the only rgb-image-setting they evaluated in [26], we show improved results over theirs, see Section 5.1. Serrà et al. [31] used a generic lossless compressor such as PNG as their general distribution model and unfortunately only reported results on the training set of the in-distribution, making their results incomparable with any other works. We show improved performance over a reimplementation (see code repository) in Section 5.1. Note that both [26] and [31] did not evaluate the cases where raw likelihoods work well, even though using the likelihood ratio may decrease performance in that case as seen in Section 5.1.

*Outlier Loss*: Hendrycks et al. [8] used a margin loss on images from a known outlier distribution (Tiny without CIFAR-images). We develop an outlier loss to improve the performance of likelihood models. It can be viewed as a combination of their idea of an outlier loss with our view of a hierarchy of distributions, and achieved an improved performance in the unsupervised setting in Section 5.3.

*Hierarchy of Features*: Regarding hierarchy of features, the closest related work investigated deep variational autoencoders that use a hierarchy of stochastic variables and found that later stochastic variables perform better at anomaly detection [20]. Furthermore, Krusinga et al. [16] developed a method to approximate probability densities from generative adversarial networks. The log densities are the sum of a change-of-volume determinant and a latent prior probability density. The latent log densities better reflected semantic similarity to the in-distribution than the full log density, however, no comparable anomaly detection results were reported. Nalisnick et al. [21] also found the same for latent vs. full log densities of Glow networks, but also did not report anomaly detection results and did not look at individual scales of Glow networks.

## 7 Discussion and Conclusion

In this work, we proposed two log-likelihood based metrics for anomaly detection, outperforming state of the art methods in the unsupervised setting and only slightly underperforming classifier-based methods in the supervised setting. For good anomaly detection performance from raw likelihoods, an additional loss (such as our outlier loss) that forces the model to assign low likelihoods for images with OOD-high-level features, e.g. wrong objects, is particularly beneficial according to empirical results. Our analysis points to a potential reason, namely that without an outlier loss, the likelihoods are almost fully determined by low-level features such as smoothness or the dominant color in an image. As such low level features are common to natural image datasets, they form a strong domain prior, presenting a difficult task to detect high-level differences between inliers and outliers, e.g., object classes.

An interesting future direction is how to best combine the hierarchy-of-distributions and hierarchy-of-features views into a single approach. Preliminary experiments freezing the first two scales of the Tiny-Glow model and only finetuning the last scale on the in-distribution have shown promise, awaiting further evaluation.

In summary, our approach shows strong anomaly detection performance particularly in the more challenging unsupervised setting and also allows a better understanding of generative-model-based anomaly detection by leveraging hierarchical views of distributions and features.

## Broader Impact

A better understanding of deep generative networks with regards to anomaly detection can help the machine learning research community in multiple ways. It allows to estimate which tasks deep generative networks may be suitable or unsuitable for when trained via maximum likelihood. With regards to that, our work helps more precisely understand the outcomes of maximum likelihood training. This more precise understanding can also help guide the design of training regimes that combine maximum likelihood training with other objectives depending on the task, if the task is unlikely to be solved by maximum likelihood training alone.

Anomaly detection in general itself has positive uses. For example, detecting anomalies in medical data can detect existing and developing medical problems earlier. Safety of machine learning systems in healthcare, autonomous driving, etc., can be improved by detecting if they are processing data that is unlike their training distribution.

Negative uses and consequences of anomaly detection can be that it allows tighter control of people by those with access to large computer and data, as they can more easily find unusual patterns deviating from the norm. For example, it may also allow health insurance companies to detect unusual behavioral patterns and associate them with higher insurance costs. Similarly repressive governments may detect unusual behavioral patterns to target tighter surveillance.

These developments may be steered in a better direction by a better public understanding and regulation for what purpose anomaly-detection machine-learning systems are developed and used.

## Acknowledgments and Disclosure of Funding

This work was partially done during an internship of Robin Tibor Schirrmeister at the Bosch Center for Artificial Intelligence. A part of this work was supported by the german Federal Ministry of Education and Research (BMBF, grant RenormalizedFlows 01IS19077C).

Yuxuan Zhou wants to thank Zhongyu Lou and Duc Tam Nguyen for the helpful discussions about the preliminary experiment results.

Robin Tibor Schirrmeister wants to thank Jan Hendrik Metzen, Polina Kirichenko, Pavel Izmailov, Manuel Watter and Dengfeng Huang for discussions and support.

## Footnotes

[2]We note that likelihood is a function of the model given the data. As the model parameters are trained to model the data density, we may abuse likelihood and density in the paper for simplicity.

[3]All results in this section are qualitatively the same with Pearson correlations.

[4]The original results of [31] are not comparable since they used the training set of the in-distribution for evaluation. We provide supplementary code that uses a publicly available CIFAR10 Glow-network with pretrained weights that roughly matches the results reported here.

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
