[Supplementary Material]

# Supplementary Material
# Understanding Anomaly Detection with Deep Invertible Networks through Hierarchies of Distributions and Features

**Robin Tibor Schirrmeister**[*]
University Medical Center Freiburg
robin.schirrmeister@uniklinik-freiburg.de

**Yuxuan Zhou**
Bosch Center for Artificial Intelligence

**Tonio Ball**
University Medical Center Freiburg

**Dan Zhang**
Bosch Center for Artificial Intelligence

This document completes the presentation of the main paper with the following:

S1: Details about Glow and PixelCNN++ architectures, and the likelihood decomposition equation Eq. (3) of Sec. 4 in the main paper;

S2: Details about the modified (local/fully connected) Glow architectures for the analysis in Sec. 2 of the main paper;

S3: Fourier-based analysis of influence of amplitude and phase on likelihoods;

S4: Details about training and evaluation of Glow and PixelCNN++, including hyperparameter choices and computing infrastructure;

S5: Details on the used datasets and dataset splits;

S6: Reasons why the results of Serrà et al. [5] are not comparable as is, and details of our reimplementation;

S7: Further quantitative results, including maximum-likelihood performance (S7.1), finetuning vs. from-scratch training (S7.2), variance over seeds (S7.3), further outlier datasets (S7.4) and different outlier losses (S7.5);

S8: Qualitative analysis of the different anomaly detection metrics;

S9: Generative vs. discriminative approach for anomaly detection.

Please also note the attached supplementary codes.

## S1  Glow and PixelCNN++ architectures

### S1.1  Glow Network architecture

Our implementation of the Glow network [2] is based on a publicly available Glow implementation [2] with one modification explained in S1.3. The multi-scale Glow network consists of three sequential scales processing representations of size $12 \times 16 \times 16$, $24 \times 8 \times 8$ and $48 \times 4 \times 4$ (channel $\times$ width $\times$ height). Each scale consists of a repeating sequence of activation normalization, invertible $1 \times 1$ convolution and affine coupling blocks, see the original paper [2] for details. Our Glow network, consistent with aforementioned public implementation, uses 32 actnorm-1 conv-affine sequences per scale.

---

[*]This work was partially done during an internship at the Bosch Center for Artificial Intelligence.
[2]https://github.com/y0ast/Glow-PyTorch/

### S1.2  PixelCNN++ architecture

We use a publicly available PixelCNN++ implementation [3] with only a single change. We reduce the number of filters used across the model from 160 to 120 for fast single-GPU training.

### S1.3  Independent $z_1$, $z_2$, $z_3$

For a multi-scale model like Glow, the overall likelihood consists of the contributions from different scales, see Eq. (3) in the main paper. In contrast to other implementations, we do not condition $z_1$ on $z_2$ or $z_2$ on $z_3$ in our Glow model as described in Sec. S1.1. Recall that Glow splits the complete latent code into per-scales latent codes $z_1$, $z_2$, $z_3$. Here, $z_1$ is the half of the output of the first scale that is not processed further. Many implementations make $z_1$ dependent of $z_2$ (and same for $z_2$ and $z_3$) as $z_1 \sim N(f(z_2), g(z_2)^2)$ with $f, g$ being small neural networks. For ease of implementation, we do not do that, instead we directly evaluate $z_1$ under a standard-normal gaussian, so $z_1 \sim N(0, 1)$.

Note that this does not fundamentally alter network expressiveness. An affine coupling layer can already implement the same computation achieved by $z_1 \sim N(f(z_2), g(z_2)^2)$. Imagine $z$ is split for the affine coupling layer into $z_1$ and $z_2$, with a coefficient network on $z_2$ used to compute the affine scale and translation coefficients $s, t$ to transform $z_1' = z_1 \odot s(z_2) + t(z_2)$. Then if $s(z_2) = -f(z_2)$ and $t(z_2) = \frac{1}{g(z_2)}$ and $z_1 \sim N(f(z_2), g(z_2)^2)$, it follows that $z_1' \sim N(0, 1)$. In other words, computing the mean and standard deviation for $z_1$ from $z_2$ is the same as normalizing $z_1$ by subtracting the mean and dividing by the standard deviation computed from $z_2$, which a regular affine coupling block can already learn.

In practice, there could still be differences due to the additional parameters, different kind of blocks used to implement $f$ and $g$, and the difference of computing the $(\log)std$ or its inverse. However, we observe no appreciable bits/dim differences between our implementation and those using the explicit conditioning step, see Section S7.1.

## S2  Local and Fully Connected Architectures

### S2.1  Local Patches

We designed our local patches experiment to train compact Glow-like models that can only process information from $8 \times 8$ patches in the original datasets. Full-sized Glow networks process the full image using three scales as written in Section S1.1. Our local Glow network instead processes local $8 \times 8$ patches using a single scale. The $32 \times 32$ input image is first cropped into 16 non-overlapping $8 \times 8$ patches. These $8 \times 8$ patches are then processed independently by a local Glow network corresponding to a single scale of the full Glow network. In other words, we treat the image as if it consists of independent $8 \times 8$ patches. Evaluating the likelihoods of these patches, their sum is the likelihood of the image assigned by the local Glow model. Note that we aimed to create a network restricted to learn a general local domain prior and not one with the best maximum-likelihood performance.

### S2.2  Fully Connected

We designed our fully-connected experiment to train fully-connected Glow networks that have a different model bias to regular convolutional Glow networks. We kept the three-scale architecture of Glow including the invertible subsampling steps at the beginning of each scale. Within each scale, the fully-connected Glow first flattens the representation, e.g. from a $12 \times 16 \times 16$ tensor per rgb-image to a 3072-sized vector in the scale 1. This vector is then processed by the usual sequence of actnorm-$1 \times 1$-affine. We next detail the processing at the scale 1, whereas the other scales follow the same design pattern. Activation normalization now processes 3072 dimensions, so has substantially more parameters. The $1 \times 1$ is now an invertible linear projection keeping the dimensionality, so a projection from 3072 dimensions to 3072 dimensions. To ensure training stability, we did not train the $1 \times 1$-projections, but kept their parameters in the randomly initialized starting state. The fully-connected affine coupling block uses a sequence of linear layer ($1536 \times 512$) - ReLU - linear layer ($512 \times 512$) - ReLU - linear layer ($512 \times 3072$) modules to compute the 1536

Figure S1: Image Mixup in frequency domain via Fourier transform. Fourier amplitudes taken from one image, phases from another and then inverted back to input space (Result). Note semantic content is more similar to phase image than to amplitude image.

translation and $1536$ scale coefficients. To allow fast single-GPU training, we reduced the number of actnorm-$1 \times 1$-affine sequences from $32$ to $8$ per scale. Similar to Section S2.1, the fully-connected network was designed to highlight the influence different model biases and not to reach the best maximum-likelihood performance.

## S3   Fourier-based Amplitude/Phase Analysis

To validate that low-level features dominate the likelihoods independent of the semantic content of the image, we create mixed images in Fourier space. Concretely, we:

1. Compute the amplitudes and phases of a batch Fourier transformed images;
2. Mix up images in their frequency domain by using one image' amplitudes and phases of the other image
3. Apply the inverse Fourier transformation to invert these mixed images to the input domain

We show examples in Figure S1. Note that the mixed images are semantically much more similar to the image the phases were extracted from. We then compare the CIFAR10-Glow likelihoods on the original images and the mixed images for SVHN and CIFAR10 images. The likelihoods of the mixed images correlate much more with the amplitude-image likelihoods (Spearman correlation $> 0.8$) than with the phase-image likelihoods (Spearman correlation $< 0.05$).

## S4   Training and Evaluation

### S4.1   Glow Training

We stayed close to the training setting of a publicly available Glow repository[4]. Namely, we use Adamax as the optimizer (learning rate $5 \cdot 10^{-4}$, weight decay $5 \cdot 10^{-5}$) and $250$ training epochs. The

training setting also includes data augmentation (translations and horizontal flipping for CIFAR10/100, only translations for SVHN, Fashion-MNIST and MNIST). These settings are the same for all experiments (from-scratch training, finetuning, with and without outlier loss, unsupervised/supervised). On 80 Million Tiny Images, we use substantially less training epochs, so that the number of batch updates is identical between Tiny and the other experiments. All datasets were preprocessed to be in the range $[-0.5, 0.5 - \frac{1}{256}]$ as is standard practice for Glow-network training, see also supplementary code.

## S4.2  PixelCNN++ Training

We stayed close to the training setting of the public PixelCNN++ repository[5]. Namely, we use Adam as the optimizer (learning rate $2 \cdot 10^{-4}$, no weight decay, negligible learning rate decay $5 \cdot 10^{-6}$ every epoch). We substantially reduced the number of epochs from 5000 to 120 to save GPU resources, and since our aim here was to show general applicability of our methods to another type of model and not to reach maximum possible performance. Consistent with the public implementation, no data augmentation is performed.

## S4.3  Numerical stabilization of the training

When training networks with outlier loss, negative infinities can appear for numerical reasons. This is actually expected as true outliers should ideally have likelihood zero and therefore log likelihood equal to negative infinity. These do not contribute to the loss in theory, but cause numerical issues. To ensure numerical stability during training, we remove examples that get assigned negative infinite likelihoods from the current minibatch. In case this would remove more than $75\%$ of the minibatch, we skip the entire minibatch. These methods are only meant to ensure numerical stability, no specific training stability methods like gradient norm clipping are used.

## S4.4  Outlier Loss Hyperparameters

As described in the main manuscript, our outlier loss is:

$$L_{\mathrm{o}} = -\lambda \cdot \log\left(\sigma\left(\frac{\log(p_{\mathrm{g}}(x_{\mathrm{g}})) - \log(p_{\mathrm{in}}(x_{\mathrm{g}}))}{T}\right)\right) = -\lambda \cdot \log\left(\frac{\sqrt[T]{p_{\mathrm{g}}(x_{\mathrm{g}})}}{\sqrt[T]{p_{\mathrm{in}}(x_{\mathrm{g}})} + \sqrt[T]{p_{\mathrm{g}}(x_{\mathrm{g}})}}\right), \quad \text{(S1)}$$

where $\sigma$ is the sigmoid function, $T$ a temperature and $\lambda$ is a weighting factor. Based on a brief manual search on CIFAR10, we use $T = 1000$ $\lambda = 6000$ as they had the highest train set anomaly detection performance while retaining stable training. PixelCNN++ training works well with the same exact values, validating the choice.

## S4.5  Evaluation Details

We use the likelihoods computed on noise-free inputs for anomaly detection with Glow networks. In practice, this means not adding dequantization noise and instead adding a constant, namely half of the dequantization interval. We found this to yield slightly better anomaly detection performance in preliminary experiments. Noise-free inputs are only used during evaluation for the anomaly detection performance. Training is done adding the standard dequantization noise introduced as in [6]. The BPD numbers reported in Section S7.1 and Table S1 also are obtained using single samples with the standard dequantization noise.

We clip log likelihoods from below by a very small number. When computing the log likelihoods from networks trained with outlier loss, negative infinities are to be expected, see Section S4.3. To include these inputs in the AUC computation, we set non-finite log-likelihoods to a very small constant ($-3000000$) before computing any log-likelihood difference.

## S4.6  Computing Infrastructure

All experiments were computed on single GPUs. Runtimes vary between $2$ to $8$ days on Nvidia Geforce RTX 2080 depending on the experiment setting (outlier loss or not, supervised/unsupervised).

## S5 Datasets

### S5.1 Dataset Splits

We use the pre-defined train/test folds on CIFAR10, CIFAR100, SVHN, Fashion-MNIST and MNIST and only train on the training fold. All results are reported on the test folds. For CelebA, we only use the first 60000 images for faster computations. We use a 1 million random subset of 80 Million Tiny Images in all of our experiments. For the Fashion-MNIST/MNIST experiments, we create a greyscaled Tiny dataset from the rgb data as $x = r \cdot 0.2989 + g \cdot 0.5870 + b \cdot 0.1140$.

### S5.2 MRI Dataset

For the MRI BRATS dataset, we also use the official train/test split. The dataset was introduced to verify the log likelihood difference on a data from a slightly different domain (medical imaging). Since our outliers defined as different modalities are not defined by the object type, we did not expect last-scale likelihood contributions $c_3(x)$ to perform as good as on the object recognition datasets. However, they still outperform raw likelihoods, by 59.2% to 53.3%.

## S6 Replication of [5]

The anomaly detection results from [5] were obtained using the training folds of the in-distribution datasets, preventing a fair comparison to our results. In written communication with Serrà et al. [5], they explained to us that the AUROC-results reported in their paper compare in-distribution training-fold examples with out-of-distribution test-fold examples. This makes a fair comparison to our results and other works impossible. In contrast and in line with standard practice, our results were obtained using the test folds of the in-distribution datasets. Unfortunately, Serrà et al. [5] are unable to provide their training code and models at the current time, so we cannot recompute their anomaly detection performance for the test fold of the in-distribution datasets. We also confirmed that depending on the training setting, the anomaly-detection AUROC values can differ substantially between the training and test fold of the in-distribution dataset.

In any case, we provide supplementary code to reproduce the method of[5] to the best of our understanding. Using a publicly available pretrained Glow-model[6], we find anomaly detection performance results similar to the ones we report for PNG as a general-distribution model.

## S7 Further Quantitative Results

### S7.1 Maximum Likelihood Performance

The maximum-likelihood-performance of our finetuned Glow networks are similar to the performance reported for from-scratch training in the original Glow paper [2]. We show the bits-per-dimension values obtained using single dequantization samples in Table S1. Note that the Glow model trained on 80 Million Tiny Images already reaches bits per dimensions on CIFAR10 and CIFAR100 close to the Glow models trained on the actual dataset (CIFAR10/CIFAR100), in line with our view that the bits per dimension are dominated by the domain prior (results also do not substantially change when including or excluding CIFAR-images from Tiny).

Our Glow-model architecture was chosen from a reimplementation of Nalisnick et al. [4] (see Section S1.1), in order to facilitate comparison of our results to other anomaly detection works. In future work, evaluating anomaly detection performance of our method with newer types of normalizing flows could be interesting.

### S7.2 Finetuning

Training Glow networks on an in-distribution dataset by finetuning a Glow network trained on Tiny substantially speeds up the training progress over training from scratch. As can be seen in Figures S3 and S2, the Glow networks reach better results after less training epochs for both maximum-likelihood

Table S1: Maximum likelihood performance in bits per dimension. Results obtained using single samples of uniform dequantization noise. Tiny is the Glow network trained on 80 Million Tiny Images. Retr refers to from-scratch training on the in-distribution dataset, Finet refers to finetuning aforementeioned Glow network trained on 80 Million Tiny Images. Note the original Glow paper [2] reached 3.35 bpd on CIFAR-10 with multi-GPU training. The Glow network and training setup we use is optimized for single-GPU training and not for maximum performance. The public implementation we originally based our implementation on (and uses the explicit conditioning step discussed in S1.3) reaches 3.39 bpd on CIFAR10.

| In-dist | Tiny | Retr | Finet |
|---------|------|------|-------|
| SVHN | 2.34 | 2.07 | 2.06 |
| CIFAR10 | 3.41 | 3.40 | 3.36 |
| CIFAR100 | 3.43 | 3.43 | 3.39 |

Figure S2: Training Curves Anomaly Detection From Scratch vs. Finetuned. Conventions as in Fig. S3. Glow networks are trained without any outlier loss. AUROC refers to AUROC computed from our log-likelihood ratio metric using another Glow-network trained on 80 Million Tiny Images. Note that the finetuned Glow networks outperform the final from-scratch trained Glow networks after less than 20% of the training epochs. Note that due to different evaluation (not noise-free) and different subsets used for intermediate results, results in this figures vary from final results in result tables.

performance and anomaly-detection performance. The improvements are strongest for CIFAR100 and weakest for SVHN, in line with CIFAR100 being the most diverse dataset and most similar to 80 Million Tiny Images.

**CIFAR10/100**          **SVHN**

Figure S3: Training Curves CIFAR10/100 and SVHN From Scratch vs. Finetuned. Transparent, thin lines indicate single-seed runs, solid, think lines indicate means over these runs. Solid horizontal lines indicate final mean performance of from-scratch trained models. Note that (i) finetuned Glow networks are better in each epoch; (ii) for CIFAR10/100 the finetuned Glow networks outperform the final from-scratch trained Glow networks after less than 20% of the training epochs and (iii) for SVHN, the finetuned Glow network outperforms the final from-scratch-trained Glow network after about 50% of the training epochs.

Figure S4: Training Curves Anomaly Detection Finetuned from Own model vs Finetuned from Tiny. Conventions as in Fig. S2. Finetuned from own model is the same as simply training twice as long on the in-distribution. At the end of training for CIFAR100, the model finetuned from Tiny still performs ~6% better on anomaly detection.

There are are still gains on log-likelihood ratio based anomaly detection for CIFAR100 when comparing finetuning a Glow a Glow network trained on Tiny to finetuning a Glow network already trained on CIFAR100 (or in other words, training the Glow Network from scratch on CIFAR100 for twice the number of epochs), see Figure S4. This is likely because the exact Tiny-model be used as the general distribution model later on, validating more similar models better cancel the model bias.

## S7.3 Result Variance across Seeds

We present the original results including standard deviation in Table S2 and provide a graphical overview over our per-seed anomaly detection results in Figure S5. Results are relatively stable across seeds.

## S7.4 Additional OOD Datasets

We report results on CelebA [3] and Tiny Imagenet [7] as additional out-of-distribution (OOD) datasets in Table S3.

## S7.5 Margin Loss vs. Outlier Loss

In our experiments, the margin-based loss introduced in [1] is less stable than our outlier loss for longer training runs, see Figure S6. Note that our results for the margin-based loss already substantially outperform the results reported for PixelCNN with a margin-based loss in [1].

## S8 Qualitative Analyses

Our different metrics (raw likelihoods, log-likelihood ratios and last-scale likelihood contributions) result in qualitatively different highest-scoring images on 80 Million Tiny Images (see Fig. S7 and S8). We take a random 120000-images subset of 80 Million Tiny Images and use our Glow network trained on CIFAR10 either with or without outlier loss to compute the metrics. Looking at the top 12 images per metric shows that using the log-likelihood ratio results in more reasonable images (closer to the inliers) than the raw likelihood, albeit mostly still simple images (see Fig. S7) and that using the Glow network trained with outlier loss results in more fitting images for all metrics (see Fig. S8).

Table S2: Anomaly detection performance summary (AUC in %). Values in parentheses are standard deviation across 3 seeds. The new term **Diff**† means to use in-distribution samples as the outliers to train Tiny-Glow, see Sec. 5.4 in main paper.

| Setting | | Unsupervised | | | Supervised | | |
|---------|----------|------------|------------|------------|------------|------------|------------|
| **In-dist** | **Out-dist** | **4x4** | **Diff** | **Diff†** | **4x4** | **Diff** | **Diff†** |
| **CIFAR10** | SVHN | 96.4 (1.4) | 98.6 (0.1) | 99.0 (0.1) | 96.1 (1.7) | 98.6 (0.1) | 99.1 (0.1) |
| | CIFAR100 | 85.4 (0.7) | 84.5 (0.6) | 86.8 (0.5) | 88.3 (0.7) | 87.4 (0.5) | 88.5 (0.3) |
| | LSUN | 95.1 (1.2) | 94.1 (1.5) | 95.8 (0.8) | 95.3 (1.1) | 94.1 (1.7) | 96.2 (0.9) |
| | Mean | 92.3 (1.0) | 92.4 (0.6) | 93.8 (0.4) | 93.3 (1.2) | 93.4 (0.8) | 94.6 (0.4) |
| **CIFAR100** | SVHN | 84.5 (2.1) | 82.2 (3.1) | 85.4 (2.1) | 89.6 (1.0) | 88.6 (0.8) | 89.4 (0.7) |
| | CIFAR10 | 61.9 (0.5) | 59.8 (0.5) | 62.5 (0.3) | 67.0 (0.6) | 64.9 (0.8) | 65.3 (0.7) |
| | LSUN | 84.6 (0.1) | 82.4 (0.3) | 85.4 (0.1) | 85.7 (0.4) | 84.3 (0.3) | 86.3 (0.2) |
| | Mean | 77.0 (0.7) | 74.8 (1.1) | 77.8 (0.6) | 80.8 (0.7) | 79.3 (0.6) | 80.3 (0.5) |
| | Mean | 84.7 (0.3) | 83.6 (0.3) | 85.8 (0.2) | 87.0 (0.9) | 86.3 (0.7) | 87.5 (0.4) |

Figure S5: Graphical Overview over Anomaly Detection Results. Markers indicate mean result over three seeds, error bars indicate standard error of that mean. Type of marker indicates type of anomaly metric (defined as before and as in Table S2). Color indicates supervised or unsupervised setting. Rows are in-distribution dataset and columns are OOD datasets. Supervised setting outperforms unsupervised setting, especially on CIFAR10 vs. CIFAR100 and vice versa. Using a general-distribution model trained with outlier loss on the in-distribution (Diff†) always outperforms general-distribution model trained without outlier loss (Diff). Relative performance of final-scale method ($4 \times 4$) compared with log-likelihood-difference methods (Diff and Diff†) varies between dataset pairs.

Table S3: Anomaly detection performance for additional OOD datasets CelebA and Tiny-Imagenet. Conventions as in Table main manuscript.

| Setting | | Unsupervised | | | Supervised | | |
|---------|----------|------------|------------|------------|------------|------------|------------|
| **In-dist** | **Out-dist** | **Raw [4x4]** | **Diff** | **Diff†** | **Raw [4x4]** | **Diff** | **Diff†** |
| **CIFAR10** | CelebA | 96.6 (1.2) | 96.1 (1.2) | 97.6 (0.5) | 96.6 (1.9) | 96.2 (2.1) | 97.8 (1.0) |
| | Tiny-Imagenet | 90.7 (0.9) | 90.6 (0.7) | 92.1 (0.4) | 91.1 (0.8) | 91.3 (0.9) | 92.7 (0.4) |
| **CIFAR100** | CelebA | 80.9 (1.3) | 76.4 (2.4) | 80.4 (1.1) | 81.9 (5.4) | 79.1 (7.2) | 81.7 (4.7) |
| | Tiny-Imagenet | 77.3 (0.5) | 77.5 (0.5) | 79.7 (0.3) | 79.5 (0.4) | 79.7 (0.5) | 80.6 (0.5) |

Figure S6: Training Curves Anomaly Detection Margin Loss vs Outlier Loss. AUROC refers to AUROC computed from our log-likelihood-difference metric using another Glow-network trained on 80 Million Tiny Images. Note Glow networks trained with margin loss experience substantial drops in anomaly detection performance in later stages of the training.

Figure S7: Most likely images from 80 Million Tiny Images for CIFAR10-Glow. 12 highest-scoring images selected according to different metrics. First row: raw likelihood, second row: log-likelihood ratio to Tiny-Glow, third row: raw last-scale $z_3$ likelihood contribution. Note that constant images attain the highest raw likelihood, showing the effect of the natural-images domain prior on the raw likelihoods. The highest-scoring log-likelihood-ratio images show a bias towards blue images and some contain actual CIFAR10 objects, namely birds and planes. Overall, the difference selects some correct images, but is still sensitive to surface features such as the global color. The last-scale results are harder to interpret, the more diverse images suggest it is slightly less affected by the domain prior of smoothness.

Figure S8: Most likely images from 80 Million Tiny Images for CIFAR10-Glow with the outlier loss. Conventions as in Figure S8. Now any of the three metrics lead to selecting mostly CIFAR10-like images, with the log-likelihood difference metric selecting more diverse images with less of a bias towards blueish images.

Table S4: Binary Classifier Anomaly Detection Results. Wide-ResNet classifier trained on 80 Million Tiny Images vs in-distribution as binary classification. We use $p_{ResNet}(y_{indist}|x)$ as our anomaly metric after training for the AUC computations.

| In-dist | OOD | AUC |
|---------|-----|-----|
| **CIFAR-10** | SVHN | 93 |
| | CIFAR-100 | 89 |
| | LSUN | 93 |
| **CIFAR-100** | SVHN | 73 |
| | CIFAR-10 | 70 |
| | LSUN | 89 |
| **SVHN** | CIFAR-10 | 100 |
| | CIFAR-100 | 100 |
| | LSUN | 100 |

## S9 Pure Discriminative Approach

As an additional baseline, we also evaluated using a purely discriminative approach. We trained a Wide-ResNet classifier to distinguish between the in-distribution and 80 Million Tiny-Images, without using any in-distribution labels. Concretely, we trained the classifier using samples of the in-distribution as the positive class and samples from 80 Million Tiny Images as the negative class in a normal supervised training setting. We use the training settings and architecture from a publicly available Wide-ResNet repository [8]. After training, we use the prediction $p_{ResNet}(y_{indist}|x)$ as our anomaly metric.

For CIFAR10/100 in-distribution, results show this baseline performs better for OOD dataset CIFAR100/10 (89% and 70% vs. 87% and 63% AUROC), similar for OOD dataset LSUN (93% and 89% vs. 96% and 86%) and worse for OOD dataset SVHN (93% and 73% vs. 99% and 85%) compared to our unsupervised generative methods (compare Table S4 to unsupervised in S2). Future work may further show what properties, advantages and disadvantages these different approaches have.

## Acknowledgments and Disclosure of Funding

Use unnumbered first level headings for the acknowledgments. All acknowledgments go at the end of the paper before the list of references. Moreover, you are required to declare funding (financial activities supporting the submitted work) and competing interests (related financial activities outside the submitted work). More information about this disclosure can be found at: `https://neurips.cc/Conferences/2020/PaperInformation/FundingDisclosure`.

Do **not** include this section in the anonymized submission, only in the final paper. You can use the `ack` environment provided in the style file to autmoatically hide this section in the anonymized submission.

## Footnotes

[3] https://github.com/pclucas14/pixel-cnn-pp/tree/16c8b2fb8f53e838d705105751e3c56536f3968a

[4]`https://github.com/y0ast/Glow-PyTorch/blob/master/train.py` (we do not use warmstart)

[5]`https://github.com/pclucas14/pixel-cnn-pp/tree/16c8b2fb8f53e838d705105751e3c56536f3968a`

[6]`https://github.com/y0ast/Glow-PyTorch`

[7] https://tiny-imagenet.herokuapp.com

[8]`https://github.com/meliketoy/wide-resnet.pytorch`, with depth=28 and widen-factor=10

[4] Eric Nalisnick, Akihiro Matsukawa, Yee Whye Teh, Dilan Gorur, and Balaji Lakshminarayanan. Do deep generative models know what they don't know? In *International Conference on Learning Representations (ICLR)*, 2019.

[5] Joan Serrà, David Álvarez, Vicenç Gómez, Olga Slizovskaia, José F. Núñez, and Jordi Luque. Input complexity and out-of-distribution detection with likelihood-based generative models. In *International Conference on Learning Representations*, 2020.

[6] L. Theis, A. van den Oord, and M. Bethge. A note on the evaluation of generative models. In *International Conference on Learning Representations (ICLR)*, 2016.