[Reviews · NeurIPS 2020]

Review 1

Summary and Contributions: ----- Update ----- I have read the author response as well as the other reviews. My major concerns have been adequately addressed in the rebuttal. Overall, I think the identification of model bias and domain prior for likelihood-based OOD are a timely contribution and valuable for the community. I raised my score to vote for accepting this submission (score: 7). ------------------ This paper investigates the recently observed phenomenon that Deep Generative Models (DGMs), here invertible normalizing flows in particular (e.g. Glow), frequently assign higher likelihood to out-of-distribution (OOD) samples (anomalies). The work investigates the hypothesis that this phenomenon is due to a combination of (a) model bias, e.g. induced via network architectures such as ConvNets, and (b) domain prior, e.g. generic low-level feature distributions such as textures for natural images. Two aspects to validate this hypothesis are analyzed, (i) a hierarchy of distributions (from generic natural images to specific image distributions), and (ii) the hierarchy of features (from low-level texture to high-level semantic features). A log-likelihood ratio model that uses two identical networks to contrast the in-distribution (e.g. CIFAR-10) to some generic background distribution (e.g. TinyImages) is proposed, inspired by the hierarchy of distributions view, which is further fine-tuned with a binary classification inspired loss (Eq. (2)). This likelihood ratio model shows an improved detection performance on common OOD benchmarks (In-/Out-of-distribution combinations on SVHN, CIFAR-10, CIFAR-100, LSUN), also over the previously suggested margin loss [3]. A second analysis following the hierarchy of features argument is presented, where the overall log-likelihood of the multi-scale Glow model is decomposed into the contributions of the various scales, showing that using the final scale likelihood (likely corresponding to higher-level, semantic features) yields significantly better OOD detection performance. ##### [1] L. Bergman and Y. Hoshen. Classification-based anomaly detection for general data. In ICLR, 2020. [2] I. Golan and R. El-Yaniv. Deep anomaly detection using geometric transformations. In NeurIPS, pages 9758–9769, 2018. [3] D. Hendrycks, M. Mazeika, and T. G. Dietterich. Deep anomaly detection with outlier exposure. In ICLR, 2019. [4] D. Hendrycks, M. Mazeika, S. Kadavath, and D. Song. Using self-supervised learning can improve model robustness and uncertainty. In NeurIPS, pages 15637–15648, 2019. [5] K. H. Kim, S. Shim, Y. Lim, J. Jeon, J. Choi, B. Kim, and A. S. Yoon. RaPP: Novelty detection with reconstruction along projection pathway. In ICLR, 2020. [6] E. Nalisnick, A. Matsukawa, Y. W. Teh, D. Gorur, and B. Lakshminarayanan. Do deep generative models know what they don’t know? In ICLR, 2018. [7] L. Ruff, R. A. Vandermeulen, N. Görnitz, L. Deecke, S. A. Siddiqui, A. Binder, E. Müller, and M. Kloft. Deep one-class classification. In ICML, pages 4393–4402, 2018. [8] L. Ruff, R. A. Vandermeulen, N. Görnitz, A. Binder, E. Müller, K.-R. Müller, and M. Kloft. Deep semi-supervised anomaly detection. In ICLR, 2020. [9] S. Zhai, Y. Cheng, W. Lu, and Z. Zhang. Deep structured energy based models for anomaly detection. In ICML, volume 48, pages 1100–1109, 2016.

Strengths: - The identification of the two influences on the issue, (a) model bias and (b) domain prior, are convincing and thoroughly justified, yielding to a manifestation of the problem in a likelihood that is dominated by generic, low-level texture features which are present in natural images in general. - This hypothesis is empirically verified in a scientifically sound manner. - The experimental evaluation demonstrates that the two proposed solutions, likelihood ratio model and likelihood decomposition corresponding to scales, result in an improved OOD detection performance.

Weaknesses: - The proposed solution to contrast the in-distribution to some generic domain dataset (e.g. TinyImages) is not new and known as Outlier Exposure [3], limiting the novelty of the work. - The two orthogonal solutions overall result in a rather mixed bag of arguments, in my mind. Following the reasoning given in the paper, I would expect the OOD detection performance to be best when the two solutions are combined, i.e. contrasting the distributions via log-likelihood ratio on the final scale. This is not the case (Table 2). Why? Am I missing something here? - I am missing a justification as to why two separate, but identical networks are used for the likelihood ratio model. Following the hierarchy arguments, these models should share similar low-level features. Why not use just one joint network? The bad performance from mixing and not matching models in Table 1 would support this view. - The analysis on scales is specific to invertible multi-scale models such as Glow. Can we draw conclusions for the more general class of DGMs (including VAEs and autoregressive models) for which this issue has been observed [6]?

Correctness: Overall, the presentation is technically sound. The empirical evaluation is scientifically rigorous. However, following the presented arguments does not fully manifest itself in the empirical results (the combination of both arguments in particular), which questions the correctness of the reasoning.

Clarity: The paper is well structured and written clearly.

Relation to Prior Work: Overall the related work is up to date and the work is well placed in the literature. Including major works from the line of research on deep anomaly detection in general [9, 7, 2, 4, 8, 1, 5], which consider other objectives such as reconstruction, one-class classification, or self-supervision could be helpful to bridge these lines.

Reproducibility: Yes

Additional Feedback: - I appreciate that you included a medical application which highlights the importance of domain-specific background distributions. - l. 58: '... refining previous likelihood-ratio-based approaches.' - Citation?


Review 2

Summary and Contributions: This paper proposes an anomaly detection method following the previous likelihood-ratio-based approaches [10,11]. The likelihood-ratio approaches calibrate the undesirably high likelihoods of out-of-distribution samples with help of a background model measuring more general, domain-wise likelihoods. The main differences of this work to the previous approaches are: 1) a model trained on 80M Tiny image dataset is employed as a background model, 2) higher-level latents of the Glow model are tested and shown to be giving a better calibrated likelihoods.

Strengths: The proposed method gives better anomaly detection performances measured by AUC compared to the previous approaches.

Weaknesses: The main weaknesses of the paper are the following two. - The proposed idea seems to be rather incremental to [10,11] and largely relying on the Glow model. - The proposed idea seems difficult to apply in other domains. To explain, employing the model trained on 80M Tiny images is a plausible idea, but the method mostly follows the framework suggested in [10, 11]. In terms of model architecture, theory and experimental settings, the method does not seem to be brining noticeable, significant progresses. Moreover, while the Tiny images are shown to be effective, finding such a general and wide dataset seems difficult for other domains. For example, can we find such a dataset for sounds? Or, if we are to detect anomalies in semiconductors by examining their images, will the Tiny image model still work given that they are quite different from the natural images? The proposed method being largely relying on the background dataset, these questions need to be answered to verify its significance. Minor: - The manuscript is difficult to read in general. It would have been more succinct and well-organized. - I could't fully understand the part regarding the local pixel differences. How should I comprehend Fig. 1 A? - Needs more detailed explanations on the prior work that the proposed method directly utilizes. In particular, the Glow model and the notion of scales in it needs to be described. - How are the correlation scores computed? Are they computed from the datasets altogether or computed from each and averaged?

Correctness: The claims, methods and experiments seem correct.

Clarity: The paper would have been more polished. It would have been more helpful to read if the contributions and hypotheses are summarized compared to the previous work.

Relation to Prior Work: The differences are well described in Sec. 6.

Reproducibility: Yes

Additional Feedback:


Review 3

Summary and Contributions: In this paper the authors propose a way to solve the issue of invertible generative models "not knowing what they don't know" in the context of anomaly detection. They to this by training an auxiliary network on a highly diverse dataset (30MTI for example) along with a network on the nominal dataset and using the difference of the likelihoods to give a score normalized to account for low level features. They include experiments demonstrating that their method is effective at improving AD using generative models and include some discussion about how their method is based on "hierarchies of features" which seems reasonable.

Strengths: Soundness: The papers method seems reasonable and their empirical evaluation seem fair. Significance: This addresses a recent issue with deep generative models and their solution seems quite effective. I would say their method is fairly significant, but not groundbreaking. Relevance: This sort of work is very appropriate for the NeurIPS community.

Weaknesses: Clarity: Section 3.1 could use some more discussion, especially to unpack (2). Soundness: Could use some other competitors, see Correctness.

Correctness: The paper should include Deep SAD (https://openreview.net/forum?id=HkgH0TEYwH) or its more recent variant (https://arxiv.org/abs/2006.00339), in Table 4 since this achieves a new SoTA on the CIFAR-10, and due to its quite different methodology from OE (Hendrycks et al. 19) might not suffer so badly from removing class information. Otherwise I find the experimental section to be fine.

Clarity: This is mostly fine. As mentioned earlier I think that (2) needs to be unpacked more. I am familiar with the PNG method in Sec. 5, but a bit more explanation is probably appropriate for the unfamiliar reader.

Relation to Prior Work: Paper's review of generative models is fine. The paper should include a more on alternative deep AD methods, particularly non-generative methods, but this isn't a dealbreaker.

Reproducibility: Yes

Additional Feedback: Despite the issues mentioned above I think the improvement towards getting a practical way to normalize out biases in likelihood scores to get a sense of what a deep generative model trained on a specific dataset doesn't know as well as general purpose generative model is significant. The intuitive explanations seem reasonable as well. Update: I’ll keep my review as is, there wasn’t much directly towards me in the response.


Review 4

Summary and Contributions: When using the likelihood provided by deep generative models for anomaly detection (or out-of-distribution detection), we often see that higher likelihoods are assigned to out-distribution samples. This paper investigated such failure from the perspectives of model bias and domain prior. Two alternative methods were proposed leveraging the views of a hierarchy of distributions and a hierarchy of features. A novel outlier loss was suggested to improve the training of the in-distribution model. Extensive experiments were performed to validate the effectiveness of the proposed methods.

Strengths: The paper investigated a failure when using the likelihood obtained by deep generative models for anomaly detection. Such investigation is crucial for the application of deep generative models. The paper tried to explain why such failure happens and provided two alternative solutions. Experiments were used to motivate the proposed views of a hierarchy of distributions and a hierarchy of features. Also, extensive experiments were performed to show the effectiveness of the proposed methods.

Weaknesses: The title of Section 2 is ``Common Low-Level Features Dominate the Model Likelihood''. However, the explanation and discuss mainly focus on one low-level feature, i.e., local pixel value differences. Besides this feature, have the authors observed similar behaviours based on other low-level features, e.g., mean over whole images. (Update based on response) The authors only explained that image mean is not low-level feature. No other examples of low-level features (as well as associated experimental observations) are provided.

Correctness: . To leverage the hierarchical view it is crucial to demonstrate what dominates the model likelihood and the paper mainly used the metric Spearman’s correlation. Nevertheless it's unclear why a higher value of Spearman’s correlation indicates that low-level features dominates the total likelihood (e.g., line 91). For example, let y = exp(x), the Spearman’s correlation is 1, which just indicates the perfect monotonic relationship between x and y. (Update based on response) The authors provided some experimental observation and intuition as argument, but did not answer reviewer's question: Spearman’s correlation is mainly used to indicate the monotonic relationship. So how and (why) can you use it to indicate dominance? . Line 203: It was claimed that for the supervised setting the proposed method reaches similar performance as the approach MSP-OE (also line 245: ``...being competitive in the supervised setting''). This description is inaccurate since based on the experiment, i.e., Table 4, MSP-OE is obviously better most of the time.

Clarity: The presentation of this paper is basically clear. Here are some comments: . Section 3.1: It would be better to briefly discuss the overall training loss when the outlier loss is incorporated. . Line 142: ``Other implementations often make z1 dependent of z2...''. Please provide the references here. . Line 170: It was mentioned that the proposed method outperforms [10] slightly (93.9% vs 93.1% AUROC). Is this result averaged over multiple seeds?

Relation to Prior Work: Inspired by the findings in [8], there are many recent works investigating the problem that the likelihood approach based on generative models could assign higher likelihoods to OOD samples. The prior work discussed in Section 6 is limited. For example, the following references are closely related: . Maaløe, Lars, et al. "Biva: A very deep hierarchy of latent variables for generative modeling." Advances in neural information processing systems. 2019. . Huang, Yujia, et al. "Out-of-Distribution Detection Using Neural Rendering Generative Models." arXiv preprint arXiv:1907.04572 (2019).

Reproducibility: Yes

Additional Feedback: The reviewer has read the response and updated the comment above. Some key questions are not well addressed. The overall score remains the same.

[Author Response · NeurIPS 2020]

We thank the reviewers for their informative feedback, indicating improved results (All), that hypotheses are "intuitive" (R4), "convincing and thoroughly justified" (R1), the investigation is "very appropriate"(R4) and "crucial" (R5) for the field, with a "scientifically rigorous", "fair" and "extensive" evaluation (R1,4,5) and 3 of 4 Rs advocating acceptance.

**Novelty (R1,R2)**: Our study indeed builds on previous work including [10,11], however, our work is not merely incremental: As R1 summarized, our manuscript contains multiple independently valuable parts. First, on the conceptual level, we develop and validate a core hypothesis about the anomaly detection failure for generative models on image datasets, explaining it as an effect of model bias and domain prior. We note post-NeurIPS-submission work [1] investigates the same question and comes to consistent conclusions, highlighting that our conceptual contributions address an important research question. Furthermore, to overcome the effects of model bias and domain prior, we use two novel viewpoints (hierarchies of distributions/features) and also derive two novel methods from them. Regarding the first method, while other log-likelihood ratio (LLR) methods have been introduced in [10,11], our LLR method differs in motivation and implementation, as our motivation results in a clear way how to choose the denominator model for the LLR, which we validate in Table 1. Our work provides not only a quantitatively but also qualitatively improved evaluation, as we describe a failure case of [11] that both [10] and [11] did not evaluate (Tab 1). Further, [11] evaluates on the *training* fold of the in-distribution, making their results incomparable due to potential overfitting (see Section 6). Further, our second, novel last-scale-likelihood method shows a completely different way to overcome model bias+domain prior with surprisingly good results, yielding more insight into the anomaly detection problem.

**Dependence on Glow Architecture (R1,R2)**: We purposely kept model architecture and experimental settings similar to prior work to facilitate comparison and focus on the novel ideas. Our LLR method works without adaptation for the autoregressive PCNN++, slightly improving over Glow (Table 1), highlighting our LLR method is not specific to Glow (note that using two separate networks allows to apply LLR to any model). The last-scale method could be evaluated on any model that allows a hierarchical decomposition of the overall likelihood (e.g., VQ-VAE-2).

**Applicability of LLR to Other Domains (R2)**: We believe our LLR method is widely applicable to many domains: (1) In many natural signal domains a suitable general dataset already exists. In this work, we already show this for typical image datasets, and additionally, without adaptation, for medical MRI images. The results of [3] on text/NLP also indicate that a suitable dataset already exists there (Wikitext-2). Since the audio domain also has domain priors like local smoothness, we think LLR to a general audio distribution will also work there, and are happy to include such experiments for the camera-ready. (2) If a suitable general dataset needs to be created, it is important to note that the general distribution does not require labels and may even profit from noisy/unclean data. Therefore there is no principal obstacle preventing collection of such data, including concatenating existing datasets.

**Complementarity of the Two Approaches (R1)**: Our LLR and our last-scale method can be viewed as two independent instances of domain prior removal methods. We do not expect them to benefit each other, as the last scale mostly contains distr.-specific information, so difference subtraction may not help. Influences on their complementarity are an interesting topic for future work, e.g.: How much domain prior info is in last scale of in-dist model? How similar is the domain prior decomposed into Glow model scales for the in-dist and the general model? Do partially joint models help?

**Two Separate Networks (R1)**: Using two networks keeps the LLR method simple and allows to train the general model once for use with different in-distr. models. Still, it is interesting future work to try a joint network (see Discussion p.8).

**Correlation Computation (R2)**: We compute the correlations of different models' likelihoods from all datasets *together* because we analyze relationships *across* datasets for anomaly detection (e.g., else, Fig 3D corr. positive).

**Spearman-Correlation Interpretation (R5)**: One motivation to use the Spearman correlation is that the rank of the likelihoods determine the resulting anomaly detection AUROC value. We redid the analysis with Pearson correlation and find almost identical results, including that likelihoods of a Glow trained on local 8x8 patches have almost perfect correlations with a Glow trained on the full image, even when trained on different datasets. Additionally, earlier scales encode low-level information and account for most of the variance of the overall likelihood (see Section 4 Fig. 3D). That shows local low-level features, beyond being correlated with the likelihood, dominate it. We also confirmed that post-hoc removing low-level features by blurring reduces likelihood correlations (will include in supplementary). Findings of a recent arXiv submission[1] also further reinforce our hypothesis.

**Overclaiming wrt MSP-OE (R5)**: We agree and would modify wording, e.g., to "slightly underperform".

**Other Low-Level Features (R5)**: For convolutional generative models, with few computational steps (our definition of low-level feature) they will extract only local features. Image mean or other global features would require many layers and thus not be low-level for Glow, we will clarify that in Section 2.

**More Extensive Related Work (R1,R4, R5)**: Thanks, we will cite and compare with the work (like BIVA) in revision.

**Fig 1A (R2)**: We will add a missing y-axis-label, i.e., the probability density (of the differences to the local mean across all patches of all images of the entire dataset). As SVHN is locally smooth, its density peaks around zero. Tiny is less smooth than SVHN, so its density is more flattened.

**Seeds (R5)**: Yes, our results are averaged over 3 seeds (same as other exp in Tab S2), will include upon revision.

**Other Minor Points (not mentioned due to space limit)**: We will address them upon revision.

## Footnotes

[1]https://arxiv.org/abs/2006.08545


[Meta-Review · NeurIPS 2020]

This work proposed two methods for likelihood-based anomaly detection: a likelihood ratio approach based on comparing in-distribution likelihood against a general background distribution likelihood, and a hierarchical approach that exploits the final scale of a Glow network. The authors did a great job in experimentally verifying their hypothesis and proposed algorithms. The idea to contrast against a general background distribution is not new per se. Nevertheless, the authors convincingly demonstrated the effectiveness of this idea in deep anomaly detection based on modern generative models. The observation that low-level features dominate the likelihood is also interesting and potentially useful for future studies. The discovery on the multi-scale difference in Glow appears to be novel and appreciated by reviewers. The experiments were thoroughly performed and nicely presented. Overall, the paper is well-written. The identification of model bias and domain prior for likelihood-based deep anomaly detection are a timely and valuable contribution for the community. Please consider taking the reviewers' comments (e.g. discussing and possibly comparing to additional related works) into your revision. The following references (and references therein) on contrastive estimation (from the pre-DL era) are also worth citing and discussing: Ingo Steinwart, Don Hush, and Clint Scovel. A classification framework for anomaly detection. Journal of Machine Learning Research, 6:211–232, 2005. Wolfgang Polonik. Measuring Mass Concentrations and Estimating Density Contour Clusters: An Excess Mass Approach. Annals of Statistics, 23:855-881, 1995.